# Pick–and–Place Trajectory Planning and Robust Adaptive Fuzzy Tracking Control for Cable–Based Gangue–Sorting Robots with Model Uncertainties and External Disturbances

**Peng Liu [1,2,\*], Haibo Tian [1], Xiangang Cao [1], Xinzhou Qiao [1], Li Gong [1], Xuechao Duan [2], Yuanying Qiu [2] and Yu Su [3]**

[1] School of Mechanical Engineering, Xi'an University of Science and Technology, Xi'an 710054, China
[2] Key Laboratory of Ministry of Education for Electronic Equipment Structure Design, Xidian University, Xi'an 710000, China
[3] School of Mechatronic Engineering, Xi'an Technological University, Xi'an 710021, China
[\*] Correspondence: liupeng@xust.edu.cn

**Abstract:** A suspended cable–based parallel robot (CBPR) composed of four cables and an end–grab is employed in a pick–and–place operation of moving target gangues (MTGs) with different shapes, sizes, and masses. This paper focuses on two special problems of pick–and–place trajectory planning and trajectory tracking control of the cable–based gangue–sorting robot in the operation space. First, the kinematic and dynamic models for the cable–based gangue–sorting robots are presented in the presence of model uncertainties and unknown external disturbances. Second, to improve the sorting accuracy and efficiency of sorting system with cable–based gangue–sorting robot, a four-phase pick–and–place trajectory planning scheme based on S-shaped acceleration/deceleration algorithm and quintic polynomial trajectory planning method is proposed, and moreover, a robust adaptive fuzzy tracking control strategy is presented against inevitable uncertainties and unknown external disturbances for trajectory tracking control of the cable–based gangue–sorting robot, where the stability of a closed-loop control scheme is proved with Lyapunov stability theory. Finally, the performances of pick–and–place trajectory planning scheme and robust adaptive tracking control strategy are evaluated through different numerical simulations within Matlab software. The simulation results show smoothness and continuity of pick–and–place trajectory for the end–grab as well as the effectiveness and efficiency to guarantee a stable and accurate pick–and–place trajectory tracking process even in the presence of various uncertainties and external disturbances. The pick–and–place trajectory generation scheme and robust adaptive tracking control strategy proposed in this paper lay the foundation for accurate sorting of MTGs with the robot.

**Keywords:** cable–based parallel robot; gangue sorting robot; pick–and–place operations; trajectory planning; tracking control; robustness

## 1. Introduction

Robotic systems have played an increasingly important role in the intelligent activity of coal mining. One practical and important area of application for robotic systems is in the intelligent identification and roboticized separation of coals and gangues with the machine vision system [1]. The separation of gangues from coals is an extremely critical link for the rational utilization of coal resources. Cable–based parallel robots (CBPRs), which have a number of desirable properties, such as simple structure, heavy payload capabilities, large workspace, low energy consumption, and so on [2–4], have been widely used in astronomical observation [5], aerial photography [6], multiple mobile cranes [7], rehabilitation and training [8], and wind tunnel experiments [9]. There has been plenty of prior work in the aspects of workspace generation and analysis [10–12],

stability evaluation and stability sensitivity analysis [1,13,14], cable tension optimal distribution [15–17], optimization design [18–20], and so on. The most remarkable characteristic of CBPRs is that it employs flexible cables instead of rigid links, while the main advantage of the robots is their high load-carrying capacity, which makes the robots suitable to be employed in pick−and−place task of the moving target gangues (MTGs). The cable−based parallel robots, according to the number of cables and degrees of freedom of the end-effector, are classified into redundant actuated CBPRs and underactuated CBPRs [21]. It should be noticed that redundant actuated CBPRs are more appropriate than underactuated ones for accurate pick−and−place operations of the heavy loads, where a high payload-to-weight ratio and a high positioning accuracy are required. Therefore, a redundantly cable−based gangue−sorting robot with an end−grab is supposed to perform pick−and−place operation of MTGs with different shapes, sizes and masses. The track, approach, pick, carry, and place operation of MTGs for the end−grab must be investigated firstly in order to accomplish the separation of coals and gangues.

Generally speaking, there inevitably are frictions between the winches and the cables that are generally time-varying and nonlinear, and therefore, the cable−based gangue−sorting robots have a complicated dynamic model, including frictional uncertainties, modeling uncertainties, and external disturbances. Similarly, the total mass of the unloaded and loaded end−grab may also change while the robot performs pick−and−place operations of MTGs. The robot in this application consists of two coupled subsystems, namely the cable−based architecture and the end−grab. Lastly, it should be pointed out that the robot controller must ensure all the cable tensions are always positive because the cables can only pull the end−grab but not push it. Consequently, the pick−and−place trajectory planning and trajectory tracking control of the robot are confronted with additional problems beyond other cable−based parallel robots.

### 1.1. Pick−and−Place Trajectory Planning

The pick−and−place trajectory planning problem of the cable−based gangue−sorting robot is a fundamental one, and it is finding a smooth and continuous trajectory from a starting position to a desired terminal position within the workspace of the robot. The aim of trajectory planning is to generate the input for the control system of the cable−based gangue−sorting robot to perform pick−and−place operation of MTGs by smooth and continuous motion of the end−grab. Thus far, there are also many publications on the trajectory planning for CBPRs. Qian [22] proposed a new trajectory planning method based on the improved quintic B-splines curves for a 3-DOF CBPR, and furthermore, the effectiveness of this method was verified through the experiments. Zhao et al. [23] presented point-to-point trajectory planning for UCPR with variable angles and height cable mast by using three algebraic methods, while the effectiveness and feasibility of the method were validated with numerical simulation and experiments. The improved rapidly exploring random tree method was proposed to address moving obstacle avoidance for CDPRs, and the suggested method was verified with the experiment [24]. Jiang et al. [25] proposed a point-to-point dynamic trajectory planning technique for reaching a series of poses with a six-DOF cable-suspended parallel robot. Hwang et al. [26] presented a scheme to suppress unwanted oscillatory motions of the payload of a four-cable-driven CDPR based on a zero-vibration input-shaping scheme, and the advantage of the proposed scheme is that it is possible to generate an oscillation-free trajectory based on a ZV input-shaping scheme, and moreover, a series of computer simulations and experiments to verify the effectiveness of the proposed method were conducted for three-dimensional motions of a CBPR with four cables. A smooth trajectory planning algorithm to enhance the smoothness of the trajectory when used in rehabilitation training was proposed for a cable-driven waist rehabilitation robot by employing an improved quintic non-uniform rational B-splines [27]. As demonstrated in [28], a novel methodology for the identification of the inertial parameters of the end-effector, based on the use of internal-dynamics equations and free-motion excitations, was proposed for

the underactuated cable-driven parallel robots, where the optimal free-motion trajectories were investigated to obtain optimal identification results. In addition, in ref. [29], a general framework for the planning of point-to-point motions that extend beyond the static workspace was presented for a six-degree-of-freedom cable-suspended parallel robot, and furthermore, the effectiveness of the proposed method was verified through simulations. Furthermore, in [30], the design and experimental validation of a novel three-DOF pendulum-like cable-driven robot capable of executing point-to-point motions by leveraging partial feedback linearization control and on-line trajectory planning based on adaptive frequency oscillators were presented. The research on the trajectory planning for CBPRs, generally speaking, mainly focuses on trajectory planning in either cable space or Cartesian space. The trajectory planning in the Cartesian space can intuitively obtain the end-effector trajectory for CBPRs. Different from other types of CBPRs, the trajectory generation of the end−grab for the pick−and−place operations of MTGs must be determined according to the movement characteristics of MTGs, the location of the gangue recovery bin, and optimal tension condition of the cables in the workspace of the robots. As a result, planning and generating the end−grab trajectory for the cable−based gangue−sorting robot in Cartesian space is the major objective under consideration in this paper, and furthermore, a four-stage trajectory planning scheme of the end−grab is proposed.

### 1.2. Pick−and−Place Trajectory Tracking Control

As mentioned above, pick−and−place trajectory tracking control of the cable−based gangue−sorting robots is a challenging problem. Moreover, it is well−known that in practice, the control system design for CBPRs with model uncertainties and external disturbances, to the extent of the authors' knowledge, is also a challenging task, and it has attracted more attention recently [31–34]. In order to reduce and eliminate the effect of nonlinear uncertainties and external disturbances on the controller of the robots, a few of approaches for CBPRs are presented, such as nonlinear adaptive control, sliding mode control, robust model predictive control, time-delay control, computed torque control, fuzzy logic control, and so on [35–39]. Izadbakhsh et al. [40] proposed a robust adaptive controller for cable-driven parallel robots subject to dynamic uncertainties, while the stability of the control system was analyzed with a Lyapunov-based method. Wang et al. [41] obtained a model-free robust adaptive control for the cable-driven parallel robots, which is composed of time-delay estimation, a new PID-NFTSM manifold, and a combined adaptive reaching law, using adaptive proportional-integral-derivative nonsingular fast terminal sliding mode. Oh et al. [42] presented an approach to design positive tension controllers for the cable-suspended parallel robots with redundant cables. Shao et al. [43] established the elastic dynamic model for the cable-driven Stewart manipulator, while the rigid-body dynamic model of the A–B rotator and the rigid Stewart manipulator was obtained in detail, and furthermore, the kinematic and dynamic vibration control strategies for the feed support system in FAST were proposed and evaluated with simulations. Duan et al. [44] presented a PID controller with base acceleration feedforward designed in the operational space of the Stewart platform based on the integrated dynamic model of the Stewart platform, and furthermore, vibration isolation and trajectory following control experiments for the cable-suspended Stewart platform was carried out. Schenk et al. [45] developed a super-twisting controller for a redundant cable-driven parallel robot to track a reference trajectory in presence of uncertainties and disturbances. Jafarlou et al. [46] investigated an adaptive fractional-order finite-time sliding mode control for the cable-suspended parallel robots in the presence of model uncertainties. The stability of the closed-loop system was analyzed with developed Lyapunov theory. Hosseini et al. [47] designed a nonlinear PD controller for cable-driven parallel robots in joint space so that the robot can track the reference trajectory quickly and accurately, while the stability of the closed-loop system was examined through Lyapunov direct method. Aghaseyedabdollah et al. [48] discussed the design of supervisory adaptive

fuzzy sliding mode control with the fuzzy PID sliding surface for a planar cable-driven parallel robot. Inel et al. [49] addressed a nonlinear continuous-time generalized predictive control for a planar cable-driven parallel robot. Zi [50] presented a three-DOF cable-driven parallel robot and its adaptive fuzzy control system design and analysis, and the simulation results showed the satisfactory performance of the proposed adaptive fuzzy control system. As a matter of fact, the dynamic model of the cable−based gangue−sorting robot is always contaminated with uncertainties such as nonlinear and time-varying parameters as well as external disturbances, and this makes the dynamic model of the cable−based gangue−sorting robots complicated, and thus, a robust controller and an adaptive fuzzy control system are required to achieve high-performance pick−and−place trajectory tracking control.

Use of the cable−based gangue−sorting robot in the pick−and−place operation of MTGs presents unique challenges: (i) the major challenge for designing a controller of the robot is that the robot may experience abrupt changes in dynamic parameters while the robot captures and places the MTGs with highly variable payload, which can cause traditional control methods to achieve poor results in practical applications; and (ii) it should be emphasized that the most important limitation of the robot is that the cables suffer from unidirectional constraints that can only be pulled and not pushed, and therefore, the cables, to perform pick−and−place operation of MTGs effectively and accurately, must be in tension in the whole workspace. For this reason, the main goal of this paper is to propose a suitable control strategy for the cable−based gangue−sorting robots. To achieve this goal, in view of the nonlinear payload variation and external disturbances of the cable−based gangue−sorting robots, a robust adaptive fuzzy tracking control, which can ensure that the cables are always in positive tension, is investigated in this paper for the high-precision tracking of the robots to efficiently and reliably perform the pick−and−place operations of the MTGs. The advantage of the proposed controller in comparison with ref. [50] is its ability to obtain the positive cable tensions along the pick−and−place trajectory in presence of model uncertainties and external disturbances, providing better tracking performance because a robust term is employed to compensate the estimation errors of the fuzzy control system.

From above, without a smooth and continuous pick−and−place trajectory and appropriate trajectory tracking control for the cable−based gangue−sorting robots, the robots might sustain serious damages, and therefore, the objective of the paper is to generate a smooth and continuous pick−and−place trajectory and to implement a robust control scheme suited for the considered pick−and−place application. As a result, the main contributions of this presented paper are summarized as follows:

i. Proposing a four-stage trajectory planning scheme for the end−grab of the cable−based gangue−sorting robot while taking account the effect of the synchronous movement of the gangues with the belt conveyor as well as the location of the gangue recovery bin;

ii. Developing a robust adaptive fuzzy control strategy in the task space to track a given trajectory for the cable−based gangue−sorting robot in the presence of model uncertainties, varying payloads, and external disturbances while guaranteeing closed-loop control system stability;

iii. Demonstrating the validity of the proposed pick−and−place trajectory planning scheme and the robust adaptive fuzzy tracking control strategy through numerical simulation.

The structure of this paper is as follows. The second section presents the kinematic and dynamic models of a cable−based gangue−sorting robot. The control system is presented in the third section. The effectiveness of the proposed pick−and−place trajectory planning scheme and robust adaptive tracking control strategy are demonstrated by simulation results in the fourth section. Finally, conclusions are drawn, and future work is presented in the fifth section.

## 2. Description and Modeling of a Cable–Based Gangue–Sorting Robot

### 2.1. Description of the Robot

In the scope of this research, the investigated gangue–sorting system with a robot, as shown in Figure 1, is composed of a cable–based gangue–sorting robot, a conveyor belt, a machine vision system, gangue recovery bin, as well as coals and gangues. The robot runs across the belt conveyor and employs four cables to drive the end–grab to move to the local neighborhood where the target gangues are located so as to complete the pick–and–place operation of the MTGs. It should be pointed that a collision-free workspace and pick–and–place trajectory can be obtained for the gangues-sorting system. According to the process and characteristics of pick–and–place operation of the target gangues, the pick–and–place trajectory of the end–grab is separated into four phases in this study, namely the starting phase, preparing phase, picking phase, and placing phase. The sorting process of the target gangues can be described in more detail as follows: in the first step, the target gangues, which move synchronously with the conveyor belt at a constant speed, will enter the visual identification area, and meanwhile, the machine vision system identifies and collects the shape and position information of the selected target gangues and transmits it to the main controller of the robot; in the next step, the target gangues move for a while to reach the picking area, where the robot performs the picking operation of the target gangues; in the final step, MTGs are placed into the gangue recovery bin, and at this point, the pick–and–place operation is completed. The robot returns to the zero position and continues to complete the pick–and–place task of other gangues. As shown in Figure 2, the proposed cable–based gangue–sorting robot consists of mechanical module and control module. The mechanical structure is composed of a frame, some pulleys, four cables and motor driven systems, and an end–grab, while the control module consists of an industrial personal computer (IPC), motion controllers, a laser tracker, encoders, and so on. It should be pointed out that the mass of the end–grab and the mass of MTGs could be available with a machine vision system by their shapes and sizes and force sensors to measure cable tensions and estimate the carried masses, while the laser tracker, which can be employed to measure 3D coordinates of the end–grab, are used in combination with the servomotor encoders to obtain the position of the end–grab. In the presence of measuring systems and equipment, the closed–loop control can be employed for the robot, which leads to performance accuracy in the pick–and–place operation of the MTGs.

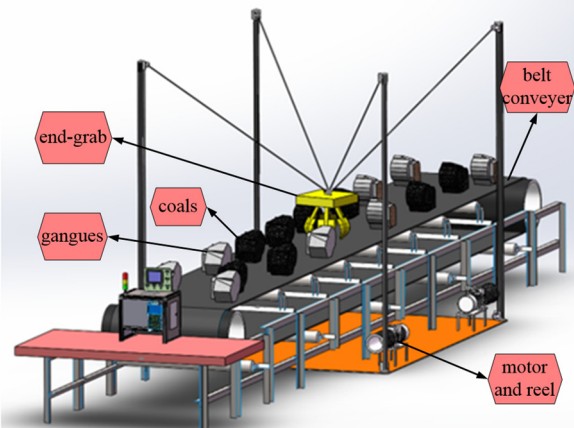

**Figure 1.** Three-dimensional CAD model of the robot.

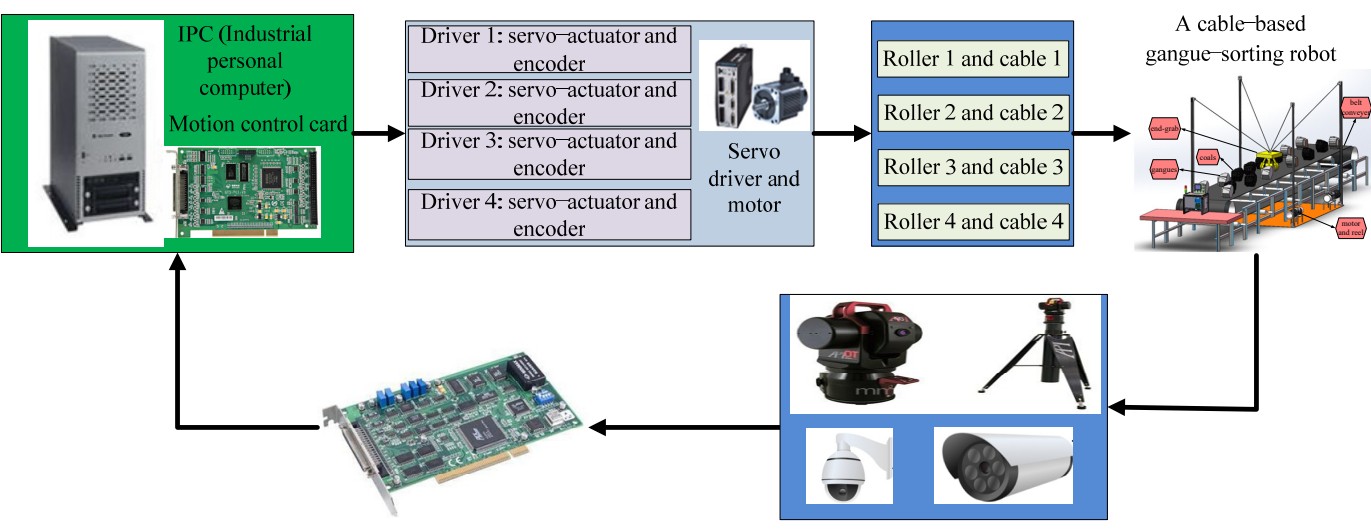

**Figure 2.** Configuration of the investigated robot.

The kinematics model of the cable−based gangue−sorting robot is a prerequisite for the dynamics model and fundamental for practical aspects such as motion trajectory planning and control system design. As a result, in this section, a full development of kinematics and dynamic models for a cable−based gangue−sorting robot is established.

### 2.2. Modeling of the Robot

As shown in Figure 3, the fixed coordinate system noted as $OXYZ$ is attached to the fixed base, where $O$ is the origin point. The structure dimensions of the robot are denoted by $a$, $b$, and $h$, respectively. The point $A_i$ ($i$ = 1, 2, 3, 4) represents the position of the fixed pulley in the coordinate system. As in ref. [1], the cable−based gangue−sorting robot can be seen by a CBPR with a point mass, and therefore, the position of the end−grab is denoted by $P$. With regard to the cable−based gangue−sorting robot, the cables, which are made of lighter materials, can be treated as a kind of massless straight line that can only sustain tension, and therefore, the length of the $i$th cable can be denoted by $L_i$. It can be easily obtained as follows:

$$L_i = P - A_i \left( i = 1, 2, 3, 4 \right) \tag{1}$$

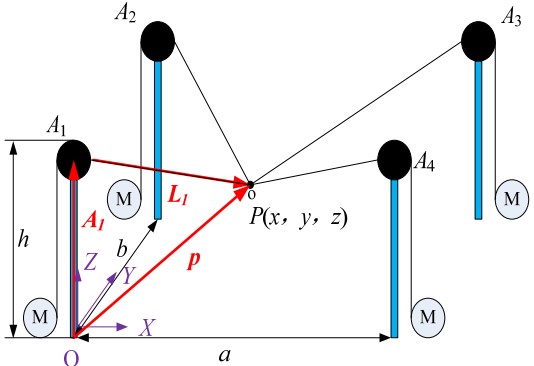

**Figure 3.** Kinematic model of the robot.

For more detail, the cable length of the four cables can be written as:

$$\begin{cases} L_1 = \sqrt{(x-x_1)^2 + (y-y_1)^2 + (z-z_1)^2} \\ L_2 = \sqrt{(x-x_2)^2 + (y-y_2)^2 + (z-z_2)^2} \\ L_3 = \sqrt{(x-x_3)^2 + (y-y_3)^2 + (z-z_3)^2} \\ L_4 = \sqrt{(x-x_4)^2 + (y-y_4)^2 + (z-z_4)^2} \end{cases} \tag{2}$$

Furthermore, the unit vector along the *i*th cable is denoted by $\boldsymbol{u}_i$, and it can be denoted by:

$$\boldsymbol{u}_i = (\boldsymbol{P} - \boldsymbol{A}_i) / L_i \tag{3}$$

A mathematical dynamic model of the cable−based gangue−sorting robot is essential for good control design and analysis to realize high-performance pick−and−place trajectory tracking control. This section presents the dynamic equations for the robot. These equations will be used later to ensure that the cable tensions remain positive during the pick−and−place operation of MTGs. There are different approaches for solving the dynamics of the robots, such as the Newton–Euler equation, Kane equation, and Lagrange equation [51–54]. In this article, the Newton–Euler equation is adopted to solve the dynamics of the robot, and thus, the dynamic equation of the robot can be expressed as:

$$m\begin{bmatrix} \ddot{x} \\ \ddot{y} \\ \ddot{z} \end{bmatrix} = \begin{bmatrix} \dfrac{x_1-x}{L_1} & \dfrac{x_2-x}{L_2} & \dfrac{x_3-x}{L_3} & \dfrac{x_4-x}{L_4} \\ \dfrac{y_1-y}{L_1} & \dfrac{y_2-y}{L_2} & \dfrac{y_2-y}{L_3} & \dfrac{y_2-y}{L_4} \\ \dfrac{h-z}{L_1} & \dfrac{h-z}{L_2} & \dfrac{h-z}{L_3} & \dfrac{h-z}{L_4} \end{bmatrix}\begin{bmatrix} T_1 \\ T_2 \\ T_3 \\ T_4 \end{bmatrix} + \begin{bmatrix} 0 \\ 0 \\ -mg \end{bmatrix} \tag{4}$$

where *m* is the mass of the end−grab; *g* is gravity acceleration; *T* is the vector consisting of all cable tensions; $\ddot{x}$, $\ddot{y}$, and $\ddot{z}$ are acceleration of the end−grab, respectively.

For the sake of simplicity, Equation (4) can be rewritten in the following matrix form:

$$\boldsymbol{M}(\boldsymbol{X})\ddot{\boldsymbol{X}} + \boldsymbol{H}(\boldsymbol{X}, \dot{\boldsymbol{X}}) = \boldsymbol{\tau} \tag{5}$$

where $\boldsymbol{M}(\boldsymbol{X}) = \begin{bmatrix} m & & \\ & m & \\ & & m \end{bmatrix}$ is the inertia matrix, which is defined as symmetric and positive; $\boldsymbol{H}(\boldsymbol{X}, \dot{\boldsymbol{X}}) = \boldsymbol{C}(\boldsymbol{X}, \dot{\boldsymbol{X}})\dot{\boldsymbol{X}} + \boldsymbol{G}(\boldsymbol{X})$ is the vector of Coriolis, centripetal, and gravity terms; $\boldsymbol{C}(\boldsymbol{X}, \dot{\boldsymbol{X}}) = \begin{bmatrix} 0 & & \\ & 0 & \\ & & 0 \end{bmatrix}$ is a nonlinear Coriolis and centripetal vector terms; $\boldsymbol{G}(\boldsymbol{X}) = \begin{bmatrix} 0 \\ 0 \\ mg \end{bmatrix}$ is gravity vector; $\boldsymbol{\tau}$ is the input torque vector; and $\boldsymbol{\tau} = \boldsymbol{J}^{\mathrm{T}}\boldsymbol{T}$ and $\boldsymbol{T} = \begin{bmatrix} T_1 \\ T_2 \\ T_3 \\ T_4 \end{bmatrix}$

are the cable tension vectors; $\boldsymbol{J}^{\mathrm{T}} = \begin{bmatrix} \dfrac{x_1 - x}{L_1} & \dfrac{x_2 - x}{L_2} & \dfrac{x_3 - x}{L_3} & \dfrac{x_4 - x}{L_4} \\ \dfrac{y_1 - y}{L_1} & \dfrac{y_2 - y}{L_2} & \dfrac{y_2 - y}{L_3} & \dfrac{y_2 - y}{L_4} \\ \dfrac{h - z}{L_1} & \dfrac{h - z}{L_2} & \dfrac{h - z}{L_3} & \dfrac{h - z}{L_4} \end{bmatrix}$ is the wrench

matrix of the robot; $\ddot{\boldsymbol{X}} = \begin{bmatrix} \ddot{x} \\ \ddot{y} \\ \ddot{z} \end{bmatrix}$ is the acceleration vector of the end−grab.

As mentioned above, when the cable−based gangue−sorting robot performs the pick−and−place operations of MTGs, its total mass of the unloaded and loaded end−grab inevitably changes. These parameter uncertainties and load variations of the end−grab will introduce disturbances to the closed-loop system for the robot and greatly affect the control performance. In practical engineering application, it is difficult to acquire complete information of the cable−based gangue−sorting robot because of parameter uncertainties and external disturbances. As a result, in the presence of the inertial parameter uncertainties and external disturbances, we can express the actual dynamic model of the robot by the following equation:

$$M(X)\ddot{X} + H(X, \dot{X}) + f + \boldsymbol{\tau}_d = \boldsymbol{\tau} \tag{6}$$

where $M(X) = M_0 + \Delta M$ and $H(X) = H_0 + \Delta H$ are the actual dynamic parameters of the robot; $M_0$ is the estimated inertia matrix and $H_0$ is the estimated Coriolis, centrifugal force, and gravity matrix, while $\Delta M$ and $\Delta H$ are dynamic modeling errors, respectively; $\boldsymbol{\tau}_d$ is the vector containing the inertial parameter uncertainties and external disturbances effects; $f$ is the viscous and Coulomb friction matrix.

Finally, we obtain the actual dynamic equation of the cable−based gangue picking robot as follows:

$$M_0(X)\ddot{X} + H_0(X, \dot{X}) + D = \boldsymbol{\tau} \tag{7}$$

where $D = \Delta M \ddot{X} + \Delta H + f + \boldsymbol{\tau}_d$ is the lumped composite disturbance including modeling errors, friction forces, and external disturbances.

It should be pointed out that the Equation (7) is a non-homogeneous linear equation, and therefore, the cable tensions may exist in infinite solutions. It must be emphasized that the equations of motion are valid only if the cables are all in tension. The suitable solution can be obtained by the cable tension optimization model with the pseudo-inverse method [16,17].

### 3. Control System

The control system of the cable−based gangue−sorting robot is responsible for: (i) planning the pick−and−place trajectory of the end−grab to track, approach, capture, carry, and place operation of MTGs and (ii) assuring realization of the designed and selected pick−and−place trajectory of the end−grab despite the parameters' uncertainty and disturbances. Pick−and−place trajectory planning can be performed while the position and dimension information of the MTG is acquired, while any controller, which is responsible for realization of the designed and selected pick−and−place trajectory of the end−grab, must work in real time. As a result, this paper proposes a new control system for the cable−based gangue−sorting robot that consists of two modules: a pick−and−place trajec-

tory planning module and a robust adaptive fuzzy tracking controller. As mentioned above, the pick−and−place trajectory planning and control of the robot can be performed in the cable space or in the Cartesian space of the end−grab. The pick−and−place trajectory planning and trajectory track control in the Cartesian space is carried out for performing the pick−and−place task of the MTGs in the paper. The position and dimension information of the MTG is obtained by using the machine vision system, and furthermore, the pick−and−place operation of the MTGs is performed autonomously after the pick−and−place trajectory is planned and generated. During execution of the pick−and−place trajectory of the end−grab, the position error of the end−grab is measured and used as a feedback for the control system.

### 3.1. Proposed Trajectory Planner

In the process of gangue sorting, the target gangue moves synchronously with the belt conveyor, and the flexible cable drives the gangue−sorting robot to complete the task of picking the gangue. The position, speed, and acceleration of the end−grab during the working process need to be set manually according to the specific requirements of the task of picking the gangue so as to achieve accurate grasping of the gangue. It is desirable to design a continuous and smooth tracking and approach trajectory for the end−grab of the cable−based gangue−sorting robot to perform the pick−and−place task of MTGs. Therefore, generation of smooth and continuous trajectory for performing the pick−and−place operation of MTGs is the major objective under consideration in this section.

### 3.1.1. Requirements for Trajectory Planning

The location and distribution of the gangues on the belt conveyor are shown in Figure 4. The geometric center of the cable−based gangue−sorting robot is located at the center line of the belt conveyor. Gangue recovery bins are arranged on both sides of the belt conveyor. Therefore, the gangues to be sorted on the belt conveyor can be considered to be distributed in zone $A$ and zone $B$, which are symmetric about the center line denoted by $b$. As a result, it is reasonable that we can take the sorting of the gangues within either zone $A$ or zone $B$ as an example to plan the pick−and−place trajectory of the end−grab for the robot, while the other side can be solved by using the symmetry relationship. We take zone $B$ as an example to illustrate the proposed pick−and−place trajectory planning scheme for the robot in this section.

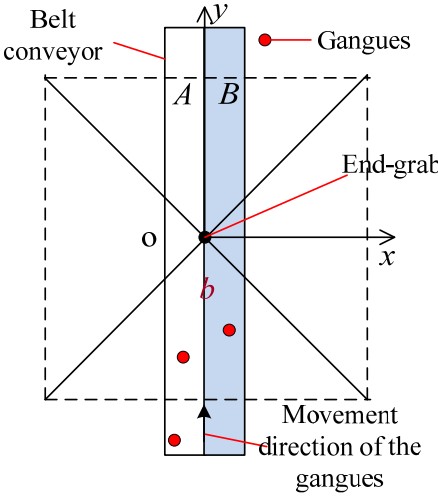

**Figure 4.** The location and distribution of the gangues.

According to the synchronous movement characteristics of MTGs with the belt conveyor, the location of the gangue recovery bin, and optimal stress condition of the

cables in the workspace of the cable−based gangue−sorting robot, we proposes a four-stage pick−and−place trajectory planning scheme for the end−grab of robot, which comprises the following four periods: the starting period, preparing period, picking period, and placing period. As a result, the pick−and−place trajectory of the end−grab must meet the following requirements:

(1) Starting period: the movement velocity of the end−grab increases from 0 to a predetermined constant speed while the robot starts for the first time. There inevitably is an acceleration stage for the end−grab, and therefore, the motion state of the end−grab should be continuous and smooth.

(2) Preparation period: in order to avoid impact during the process of picking the target gangue, the end−grab and the target gangue to be grabbed should be in a relatively static state. Therefore, there needs to be a constant linear motion along the movement direction of the target gangue.

(3) Picking period: no impact can occur during the operation of carrying the picked gangue after the target gangue is captured by the end−grab. In addition, in order to ensure that the captured gangue can fall into the gangue bin at a fixed direction and speed, the end−grab should also have a uniform linear motion at this stage.

(4) Placing period: in order to avoid repeated acceleration leading to a discontinuous trajectory for the end−grab, the end−grab directly enters stage (2) to perform the pick−and−place operation of the next target gangue after the current picked gangue is placed in the gangue recovery bin.

In addition, each trajectory for the four periods mentioned above should be smoothly connected to the next one to avoid the motion state discontinuity in the neighborhood of the trajectory transition point, which can lead to dynamic impact.

### 3.1.2. Trajectory Planning Scheme

(1) Determination of the end−grab position and zero position

As shown in Figure 5, considering the unidirectional characteristics of the cables, optimal tension condition of the cables, and the workspace where the end−grab is located, the tension of each cable is equal to each other while the end−grab is located at the geometric center of the workspace, so the ideal position of the end−grab for grabbing the target gangue should be located on the vertical axis of the workspace. In addition, the target gangue moves along the positive direction of *y*-axis. Considering the tension condition of the cable, a certain position along the *x*-axis can be selected as the most appropriate grabbing point, and thus, point *E* is selected as the grabbing point of the target gangue. Then, the straight line is perpendicular to the movement direction of the target gangue and the belt through *E* point, which can be obtained, and the straight line crosses the belt at points *Q* and *R* on both sides. As a result, the grabbing point of the target gangue must be on the *QR* while the position coordinate of *x*-direction deviates from the center line *b*. As requirement (2) states, the end−grab and the target gangue to be grabbed should be in a relatively static state, and therefore, the zero point the end−grab should be ahead of the grab point to achieve the synchronous movement of the end−grab and the target gangue. As a result, point *C* is chosen as the zero point of the end−grab for the cable−based gangue−sorting robot.

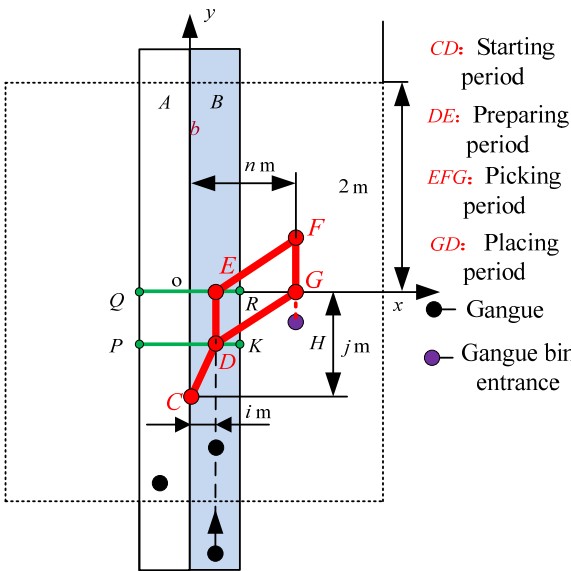

**Figure 5.** Pick−and−place trajectory planning scheme.

(2) Determination of the starting period and preparation period

According to the requirement (2) of trajectory planning, point *E* is the terminal point of the uniform motion in a straight line of the end−grab, while the starting point of the uniform linear motion of the end−grab should be reasonably selected. Point *D* is selected as the starting point of the uniform linear motion. Then the straight line is perpendicular to the movement direction of the target gangue and the belt through *D* point, which can be obtained, and the straight line crosses the belt at points *P* and *K* on both sides. As a result, the starting point of the uniform linear motion is on *PK* no matter where the target gangues are located. Therefore, the *CD* segment is the starting period while the *DE* segment is the preparation period. It should be noted that the starting section *CD* and preparing section *DE* coincide on the same straight line when the target gangue is located at the center line *b* of the conveyor belt, and thus, the end−grab does not move in the *x*-direction.

(3) Determination of the picking period

According to the requirement (3) of the trajectory planning, the end−grab should also move in a straight line at a constant speed at this stage. The cable tensions will change while the target gangue is placed, so the reasonable point should be selected as the placing point of the target gangues. Considering the fact that the cable tensions are relatively reasonable when the end−grab is on the *x*-axis, point *G* is selected as the terminal point of the straight line. Connecting the line segment *HG* and the starting point *F* of the uniform linear motion must happen on the extension line of *HG* along the positive direction of the *y*−axis, and the position of point *F* can be determined while the operation time of the end−grab on *FG* is consistent with *DE*. As a result, *EFG* section is the picking period when the target gangue is carried by the end−grab from point *E* to point *G*.

(4) Determination of the placing period

According to the requirement (4) of trajectory planning, the end−grab will enter the placing period after the target gangue is placed by the end−grab at *G* point to avoid repeatedly returning to zero to accelerate. The end−grab can directly transition to the preparation section after the current picked gangue is placed in order to avoid repeated acceleration, leading to a discontinuous trajectory for the end−grab. Thus, *GD* is the placing period of the pick−and−place trajectory for the end−grab.

To sum up, as shown in Figure 5, point *C* is selected as the zero position of the end−grab; point *D* and point *F* are the starting points of the uniform linear motion of the end grab, respectively, while point *E* is the grabbing point of the target gangues, and

point *G* is the placing point of the target gangues. The *CD* segment, the *DE* segment, the *EFD* segment, and the *GD* segment are the starting period, the preparation period, the picking period, and the gangue-placing period, respectively. The four periods above shall be smoothly connected to ensure that the end−grab will not be impacted during the process of the pick−and−place operation of target gangues.

### 3.1.3. Implementation Methods of Each Trajectory for the Four Periods

(1) S-shaped acceleration/deceleration algorithm

The S-shaped acceleration/deceleration algorithm is optimized on the basis of T-shaped velocity programming. The planned trajectory, velocity, and acceleration are continuous, which can ensure smooth acceleration and deceleration of the end-effector without impact [55,56]. There are four types of S-shaped velocity curve planning: seven-stage, six-stage, five−stage, and four-stage, respectively. Considering the fact that the motion planning of the end−grab meets the conditions of five−stage planning, the five−stage S-shaped velocity curve planning method is introduced here, whose velocity and acceleration curves are shown in Figure 6. As shown in Figure 6, $v_{\max}$ represents the maximum planned speed; $t_i$ represents the time; $T_i = T_{i+1} - T_i$ is the time of each segment; $a_{\max}$ represents the maximum acceleration; and *J* represents the jerk. As a result, the relationship between the total displacement of the end−grab *s*, displacement of the acceleration stage $s_1$, the jerk *J*, and acceleration of the end−grab $a_{\max}$ is as follows:

$$s_1 = \frac{Jv_{\max}^2 + v_{\max}a_{\max}^2}{2Ja_{\max}} \tag{8}$$

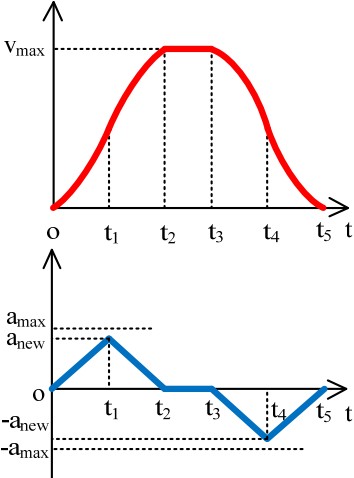

**Figure 6.** Velocity and acceleration curves of the five−stage.

If the above parameters meet $v_{\max} \le \dfrac{a_{\max}^2}{J}$ and $s > 2v_{\max}\sqrt{\dfrac{v_{\max}}{J}}$ in the given range of the total displacement of the end−grab *s*, the acceleration cannot reach the maximum value, but the speed can reach the maximum value, and in this case, the maximum acceleration needs to be adjusted as follows:

$$a_{new} = \sqrt{v_{\max}J} \tag{9}$$

Further, the time of each segment can be obtained as follows:

$$T_1 = T_2 = T_4 = T_5 = \frac{a_{new}}{J} \tag{10}$$

$$T_3 = \frac{s - 2v_{\max}\sqrt{\dfrac{v_{\max}}{J}}}{v_{\max}} = \frac{s}{v_{\max}} - 2\sqrt{\frac{v_{\max}}{J}} \tag{11}$$

In this paper, this method is only applied to the acceleration section of *y*-direction, and the expressions of its speed and displacement acceleration section are as follows:

$$v(t) = \begin{cases} \dfrac{1}{2}Jt^2 & 0 < t < T_1 \\ \dfrac{1}{2}JT_1^2 + JT_1 t - \dfrac{1}{2}Jt^2 & T_1 < t < T_2 \end{cases} \tag{12}$$

$$s(t) = \begin{cases} \dfrac{1}{6}Jt^3 & 0 < t < T_1 \\ \dfrac{1}{6}JT_1^3 + \dfrac{1}{2}JT^2{}_1 t + \dfrac{1}{2}JT_1 t^2 - \dfrac{1}{6}Jt^3 & T_1 < t < T_2 \end{cases} \tag{13}$$

(2) Quintic polynomials

The four trajectories of the four periods should be smoothly connected to each other. Therefore, the sharp points, which will cause local discontinuity of the speed curve and impact of the end−grab, are avoided in the trajectory. The quintic polynomial trajectory planning, which can ensure the continuity up to the acceleration level, is employed to plan the pick−and−place trajectory of the end−grab [57]. The position, velocity, and acceleration equations of quintic polynomial trajectory planning can be expressed as:

$$\begin{bmatrix} q(0) \\ \dot{q}(0) \\ \ddot{q}(0) \\ q(t) \\ \dot{q}(t) \\ \ddot{q}(t) \end{bmatrix} = \begin{bmatrix} 1 & 0 & 0 & 0 & 0 & 0 \\ 0 & 1 & 0 & 0 & 0 & 0 \\ 0 & 0 & 2 & 0 & 0 & 0 \\ 1 & t & t^2 & t^3 & t^4 & t^5 \\ 0 & 1 & t & t^2 & t^3 & t^4 \\ 0 & 0 & 1 & t & t^2 & t^3 \end{bmatrix} \begin{bmatrix} a_0 \\ a_1 \\ a_2 \\ a_3 \\ a_4 \\ a_5 \end{bmatrix} \tag{14}$$

where $q = \left[q(0), \dot{q}(0), \ddot{q}(0), q(t), \dot{q}(t), \ddot{q}(t)\right]^{\mathrm{T}}$ is a vector composed of generalized coordinates, generalized velocity, and generalized acceleration from the starting point to the terminal point. $a_i$ represents the coefficients of the quintic polynomial.

From above, the trajectory planning methods adopted by each period are shown in Table 1.

**Table 1.** Trajectory planning method of each section.

| Periods | Direction | Used Trajectory Planning Method |
|---|---|---|
| | *x* | Quintic polynomials |
| Starting period *CD* | *y* | S-shaped acceleration/deceleration algorithm |
| | *z* | No displacement |
| | *x* | No displacement |
| Preparation period *DE* | *y* | S-shaped acceleration/deceleration |
| | *z* | No displacement |
| | *x* | Quintic polynomials |
| Picking period *EF* | *y* | Quintic polynomials |
| | *z* | Quintic polynomials |
| | *x* | No displacement |
| Picking period *FG* | *y* | S-shaped acceleration/deceleration |
| | *z* | Quintic polynomials |
| | *x* | Quintic polynomials |
| Placing period *GD* | *y* | Quintic polynomials |
| | *z* | No displacement |

*3.2. Proposed Controller*

For a cable–based gangue–sorting robot, the external disturbances, frictional terms, internal uncertainties, and payload variation are considered as a composite disturbance, which mostly affects the control performance and causes inaccuracy in the control process. The main objective of control for the cable–based gangue–sorting robot is to achieve high-performance pick–and–place trajectory tracking accuracy in the pick–and–place operation of MTGs. As a result, considering the structural characteristics and control difficulties of the cable–based gangue–sorting robot, in the presented control system, a robust adaptive fuzzy tracking control is used for assuring realization of the selected trajectory of the end–grab to perform the pick–and–place operation of MTGs for the cable–based gangue–sorting robot. The uncertainties of the control system are adaptively compensated by fuzzy control system, while a robust term is employed to compensate the estimation errors of the fuzzy control system. Meanwhile, the stability of the whole closed-loop system is guaranteed.

As an intelligent control method, fuzzy control is based on fuzzy logic inference, and the microcomputer control method has also been widely applied to generate auxiliary joint torques to compensate these uncertainties [58,59]. Fuzzy logic system can be employed to approximate the unknown nonlinear functions as well as external disturbances. It is especially suitable for the control of nonlinear, time-varying systems, such as robotics. The approximation characteristics of the fuzzy logic system are used to compensate for the composite disturbance for the cable–based gangue–sorting robot. A robust term is designed to eliminate the estimation errors and external disturbance of fuzzy logic system.

A multi-input and multi-output fuzzy logic system performs mapping from fuzzy sets in $U \in R^n$ to fuzzy sets in $V \in R^m$, based on the fuzzy IF–THEN rules. The output of a multi-input and multi-output fuzzy logic system with center-average defuzzifier, product inference, and singleton fuzzifier takes the following form:

$$y_j = \frac{\sum_{l=1}^{M} \bar{y}_j^l \left( \prod_{i=1}^{n} \mu_{A_i^l}(x_i) \right)}{\sum_{l}^{M} \left( \prod_{i=1}^{n} \mu_{A_i^l}(x_i) \right)}, (j = 1, 2, ..., m) \tag{15}$$

where $\bar{y}_j^l$ is a value at which the membership function for output fuzzy set reaches its maximum; $\mu_{A_i^l}(x_i)$ is the membership function of the linguistic variable $x_i$ and can be defined as follows:

$$\mu_{A_i^l}(x_i) = \exp\left( \frac{-(x_i - \bar{x}_i^l)}{\sigma^2} \right)^2 \tag{16}$$

where $\bar{x}_i^l$ and $\sigma$ are the mean and the deviation of the Gaussian membership function, respectively.

The fuzzy basis function can be defined as:

$$\varepsilon_l(x) = \frac{\prod_{i=1}^{n} \mu_{A_i^l}(x_i)}{\sum_{l}^{M} \left( \prod_{i=1}^{n} \mu_{A_i^l}(x_i) \right)} \tag{17}$$

As a result, Equation (15) can be rewritten as follows:

$$y_j = \Theta_j^T \varepsilon(x) \tag{18}$$

where $\varepsilon(x)=[e_1(x),e_2(x),...,e_M(x)]^{\mathrm{T}}$ is the fuzzy basis function vector, while $\Theta_j=[\bar{y}_j^1,\bar{y}_j^2,...,\bar{y}_j^M]^{\mathrm{T}}$ is the center of the fuzzy subset.

Furthermore, the overall output of a MIMO fuzzy logic system can be represented as:

$$y = \Theta^{\mathrm{T}}\varepsilon(x) \tag{19}$$

The control problem for the cable−based gangue−sorting robot is to design a controller to compute the cable tensions $T$ applied to the end−grab such that the end−grab positions $X$ tend asymptotically toward the constant desired end−grab positions $X_d$. Therefore, the tracking error is defined as:

$$e = X(t) - X_{\mathrm{d}}(t) \tag{20}$$

where $X_{\mathrm{d}}(t)$ is the desired trajectory of the end−grab, while $X(t)$ is the actual trajectory of the end−grab.

Moreover, the sliding surface $s$ is defined as follows:

$$s = \dot{e} + \Lambda e \tag{21}$$

where $\Lambda$ is a positive definite parameter matrix.

The reference tracking velocity can be defined as follows:

$$\dot{X}_r(t) = \dot{X}_{\mathrm{d}}(t) - \Lambda X(t) \tag{22}$$

The proposed controller, which can counteract the external disturbances, frictional terms, and internal uncertainties, can be expressed by the following equation:

$$
\begin{aligned}
&T = T_n + T_h \\
&T_n = \left(J^{\mathrm{T}}\right)^+\left(M(X)\ddot{X}_r + H\left(X,\dot{X}\right) + \hat{F}(X,\dot{X},\ddot{X}|\tilde{\Theta}) - K_D s - W\mathrm{sgn}(s)\right) \\
&T_h = \left(I - \left(J^{\mathrm{T}}\right)^+ J^{\mathrm{T}}\right)\lambda \\[4pt]
&\hat{F}(X,\dot{X},\ddot{X}|\tilde{\Theta}) = \Theta_i^{\mathrm{T}}\varepsilon(X,\dot{X},\ddot{X})
\end{aligned} \tag{23}
$$

in which $T_n$ is the special solution to the vector $T$, while $T_h$ is the homogeneous solution to the vector $T$ and $J^{\mathrm{T}}T_h = 0$; $\left(\bullet\right)^+$ denotes the pseudo inverse; $K_D = diag(K_i), K_i > 0$ ; $\lambda$ is an arbitrary scalar; $W = diag[\omega_{M_1},\omega_{M_2},....,\omega_{M_n}], \omega_{M_i} \geq |\omega_i|, i=1,2,...,n$ ; $\hat{F}(X,\dot{X},\ddot{X}|\tilde{\Theta})$ is the fuzzy logic compensation control for the lumped composite disturbance, and it is represented as

$$
\hat{F}(X,\dot{X},\ddot{X}|\tilde{\Theta})=\begin{bmatrix} \hat{F}(X,\dot{X},\ddot{X}|\tilde{\Theta}_1) \\ \hat{F}(X,\dot{X},\ddot{X}|\tilde{\Theta}_2) \\ \hat{F}(X,\dot{X},\ddot{X}|\tilde{\Theta}_3) \end{bmatrix}=\begin{bmatrix} \Theta_1^{\mathrm{T}}\varepsilon(X,\dot{X},\ddot{X}) \\ \Theta_2^{\mathrm{T}}\varepsilon(X,\dot{X},\ddot{X}) \\ \Theta_3^{\mathrm{T}}\varepsilon(X,\dot{X},\ddot{X}) \end{bmatrix}.
$$

As stated in the introduction, the proposed robust adaptive fuzzy tracking controller has an advantage over the one in ref. [50]. It is assumed that the cables, however, are ideally massless and nonelastic elements in this paper. It should be pointed out that the cables may behave as elastic elements in practice. This elasticity of the cables inevitably causes unwanted vibrations, leading to degradation of the positioning accuracy of the end−grab for the cable−based gangue−sorting robot. For this reason, the proposed controller for the cable−based gangue−sorting robot should efficiently dampen the vibrations of the cables, leading to enhancement of the motion accuracy of the end−grab [60].

Future research will be dedicated to investigate the effect of the cable vibration on the controller for the robots using the singular perturbation theory.

By using Lyapunov theory, the stability of the cable−based gangue−sorting robot according the dynamics in the presence of the disturbances expressed in Equation (7) with the robust adaptive fuzzy tracking control in Equation (23) is proven. As a result, the Lyapunov function candidate is defined as:

$$V(t) = \frac{1}{2}\left( \boldsymbol{s}^T \boldsymbol{M} \boldsymbol{s} + \sum_{i=1}^{n} \tilde{\Theta}_i^T \Gamma_i \tilde{\Theta}_i \right) \tag{24}$$

$$
\begin{aligned}
\dot{V}(t) \quad &= -\boldsymbol{s}^T(\boldsymbol{M}\ddot{\boldsymbol{X}}_r + \boldsymbol{H}\dot{\boldsymbol{X}}_r + \boldsymbol{F}(\boldsymbol{X},\dot{\boldsymbol{X}},\ddot{\boldsymbol{X}}|\tilde{\Theta}) - \boldsymbol{J}^T\boldsymbol{T}) + \sum_{i=1}^{n} \tilde{\Theta}_i^T \Gamma_i \dot{\tilde{\Theta}}_i \\[2mm]
&= -\boldsymbol{s}^T(\boldsymbol{F}(\boldsymbol{X},\dot{\boldsymbol{X}},\ddot{\boldsymbol{X}}) - \hat{\boldsymbol{F}}(\boldsymbol{X},\dot{\boldsymbol{X}},\ddot{\boldsymbol{X}}|\tilde{\Theta}) + \boldsymbol{K}_D\boldsymbol{s} + \boldsymbol{W}\mathrm{sgn}(\boldsymbol{s})) + \sum_{i=1}^{n} \tilde{\Theta}_i^T \Gamma_i \dot{\tilde{\Theta}}_i \\[2mm]
&= -\boldsymbol{s}^T(\boldsymbol{F}(\boldsymbol{X},\dot{\boldsymbol{X}},\ddot{\boldsymbol{X}}) - \hat{\boldsymbol{F}}(\boldsymbol{X},\dot{\boldsymbol{X}},\ddot{\boldsymbol{X}}|\Theta) + \hat{\boldsymbol{F}}(\boldsymbol{X},\dot{\boldsymbol{X}},\ddot{\boldsymbol{X}}|\tilde{\Theta}^*) - \hat{\boldsymbol{F}}(\boldsymbol{X},\dot{\boldsymbol{X}},\ddot{\boldsymbol{X}}|\tilde{\Theta}^*) + \boldsymbol{K}_D\boldsymbol{s} + \boldsymbol{W}\mathrm{sgn}(\boldsymbol{s})) + \sum_{i=1}^{n} \tilde{\Theta}_i^T \Gamma_i \dot{\tilde{\Theta}}_i \\[2mm]
&= -\boldsymbol{s}^T(\tilde{\Theta}^T \varepsilon(\boldsymbol{X},\dot{\boldsymbol{X}},\ddot{\boldsymbol{X}}) + \omega + \boldsymbol{K}_D\boldsymbol{s} + \boldsymbol{W}\mathrm{sgn}(\boldsymbol{s})) + \sum_{i=1}^{n} \tilde{\Theta}_i^T \Gamma_i \dot{\tilde{\Theta}}_i \\[2mm]
&= -\boldsymbol{s}^T\boldsymbol{K}_D\boldsymbol{s} - \boldsymbol{s}^T\omega - \boldsymbol{s}^T\boldsymbol{W}\mathrm{sgn}(\boldsymbol{s}) + \sum_{i=1}^{n} (\tilde{\Theta}_i^T \Gamma_i \dot{\tilde{\Theta}}_i - s_i \tilde{\Theta} \varepsilon(\boldsymbol{X},\dot{\boldsymbol{X}},\ddot{\boldsymbol{X}}))
\end{aligned}
\tag{25}
$$

where $\omega$ is the fuzzy approximation error; $\tilde{\Theta}_i = \Theta_i^* - \Theta_i$; $\varepsilon(\boldsymbol{X},\dot{\boldsymbol{X}},\ddot{\boldsymbol{X}})$ is fuzzy basis function.

Moreover, the adaptive law based on Equation (25) is defined as:

$$\dot{\tilde{\Theta}}_i = -\Gamma_i^{-1} s_i \varepsilon(\boldsymbol{X},\dot{\boldsymbol{X}},\ddot{\boldsymbol{X}}) \tag{26}$$

Consequently, the following equation can be obtained:

$$\dot{V}(t) \leq -\boldsymbol{s}^T \boldsymbol{K}_D \boldsymbol{s} \leq 0 \tag{27}$$

It can be seen from Equations (24) and (27) that, since function **V** is a positive definite function, and $\dot{V}$ is a negative definite function, the cable−based gangue−sorting robot in Equation (7) controlled by the proposed robust adaptive fuzzy tracking controller in Equation (23) is globally asymptotically stable with respect to **s** and $\Theta$ based on the Lyapunov method. It means that $\lim_{t\to\infty}\boldsymbol{s} = 0$, and thus, this is equivalent to $\lim_{t\to\infty}\boldsymbol{e} = 0 \Rightarrow \lim_{t\to\infty}\boldsymbol{X} = \lim_{t\to\infty}\boldsymbol{X}_d$. As a result, the control object for the end−grab to track the scheduled trajectory to perform the pick−and−place operation of MTGs can be realized.

## 4. Simulation Study

### 4.1. Generation of the Proposed Pick−and−Place Trajectory

The end−grab of the robot is requested to start from the state of rest, merge into four consecutive trajectories in sequence to perform the pick−and−place operation of the current target gangue, and then go back to the point *D* to begin the pick−and−place operation of the next gangue. Assuming that the distance the target gangue is from the centerline of the belt conveyor is 0.25 m, the pick−and−place trajectory of the end−grab is achieved in this example. As is shown in Figure 3, some parameters of the cable−based gangue−sorting robot are given as follows: *a* = 4 m, *b* = 4 m, and *h* = 3 m. In order to ensure the robustness and the stability of the cable−based gangue−sorting robot, in this paper, all positions of the end−grab from a planned trajectory are required to be completely within

the stability workspace, which refers to the set of points meeting certain stability requirements of the end−grab [61]. The stability workspace is composed of the positions of the end−grab with specified stability performance index, which can be calculated using the position and cable tension influencing factors, and furthermore, it can be obtained by the stability workspace generation algorithm as described in the previously published paper, ref. [61], by the author.

Figure 7 shows the pick−and−place trajectory of the end−grab and the positional relationship between the trajectory and the stability workspace. It should be noted that the pick−and−place trajectory is obtained while the distance from the position of the MTG to be grabbed to the center line $b$, denoted by $j$, is 0.25 m. It is observed from Figure 7a that the pick−and−place trajectory consists of four periods, and furthermore, each period's connection with each other is smooth, and thus, this leads to no impact for the movement of the end−grab. In addition, it can be seen from the Figure 7b that the pick−and−place trajectory of the end−grab is located completely within the stability workspace, and this can ensure the stability of the end−grab. Indeed, it should be pointed out that all of the positions when the end−grab is located within the surface meet the specified stability requirement. Moreover, its colors are worthy of note, as they represent the elevation of the positions for the end−grab along $z$−direction. As can be seen from Figure 7c,d, the velocities and accelerations of the end−grab along the proposed pick−and−place trajectory are continuous.

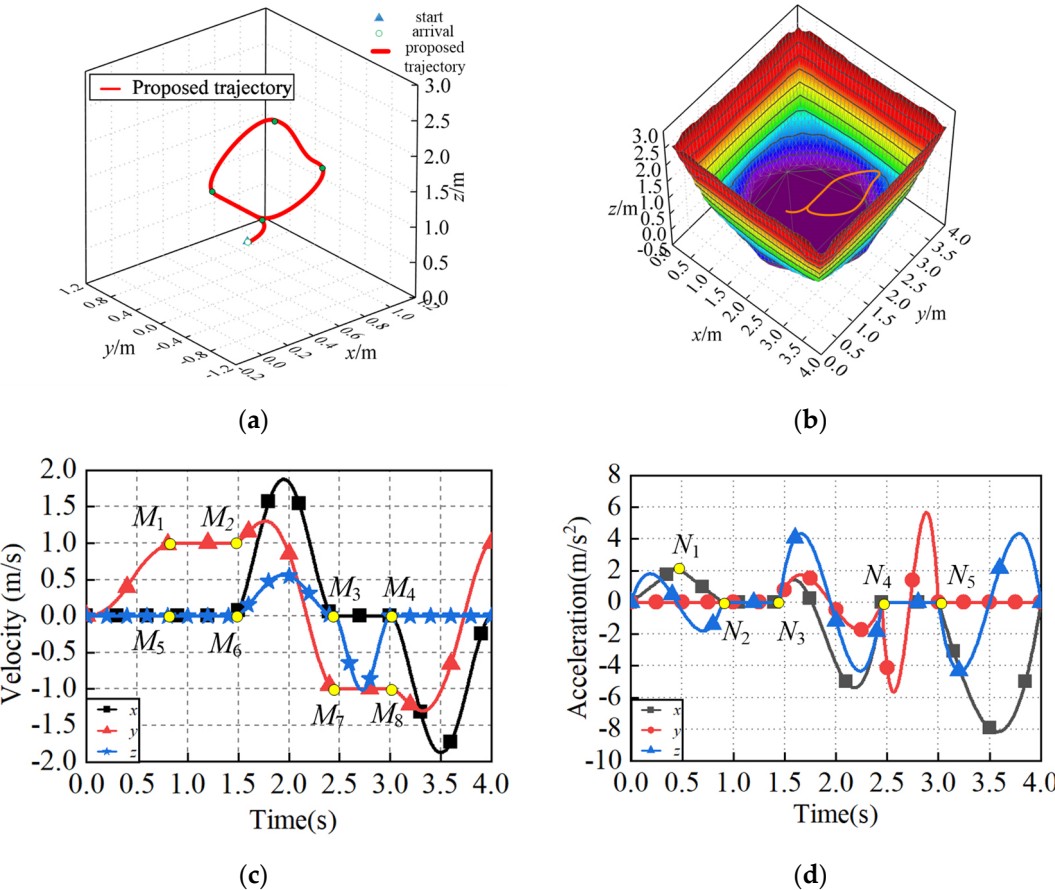

**Figure 7.** Pick−and−place trajectory of the end−grab. (**a**) Pick−and−place trajectory; (**b**) trajectory within the stability workspace; (**c**) velocities of the end−grab; (**d**) accelerations of end−grab.

In addition, the proposed strategy can generate a smooth and continuous pick−and−place trajectory while the MTG is located at an arbitrary position of the belt, such as the center line $b$, namely $j = 0$, shown in Figure 8. Comparing Figures 7a and 8, it

is clear that the trajectories of the starting period *CD* are slightly different from each other, and in more detail, there is no displacement of the end−grab along the *x*−direction for the trajectory of the starting period *CD* in Figure 8. This is because the pick−and−place position of the MTG is on the center line *b*.

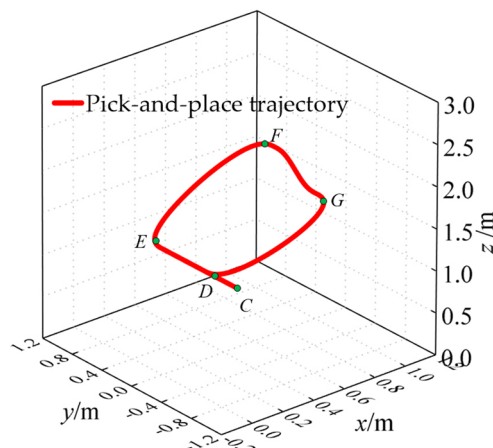

**Figure 8.** Pick−and−place trajectory of the end−grab while *j* = 0.

Figure 9 displays the length of the four cables. From the simulation results, it may be concluded that cable 1 and cable 3, cable 2, and cable 4 show opposite trends in the whole trajectory curve because of the symmetrical geometric relationship between the cables, and moreover, the length of the four cables change smoothly and continuously.

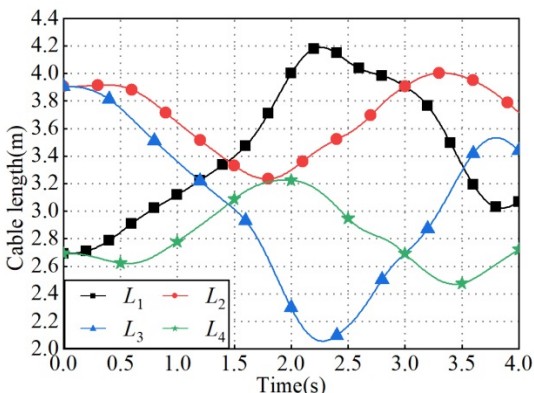

**Figure 9.** Length of the four cables.

### 4.2. Control System Validation

In this section, the proposed pick−and−place trajectory control strategies are evaluated and compared with MATLAB software. Two motion trajectories for the end−grab, a spatial circle, and the proposed pick−and−place trajectory generated in this paper are considered in this section to illustrate the efficiency of the designed controller.

To illustrate the effectiveness of the proposed robust adaptive fuzzy tracking controller, we compared it with the fuzzy controller presented in ref. [50]. The controller mentioned above was modified to be implementable for the cable−based gangue−sorting robot considered in this paper. Thus, it can be expressed as:

$$\boldsymbol{\tau} = \boldsymbol{M}(X)\ddot{X}_r + \boldsymbol{H}\left(X, \dot{X}\right) + \hat{\boldsymbol{f}}\left(X, \dot{X} \middle| \tilde{\Theta}\right) - \boldsymbol{K}_D \boldsymbol{s} \tag{28}$$

in which

$$\hat{f}(X,\dot{X}|\tilde{\Theta})=\begin{bmatrix}\hat{f}(X,\dot{X}|\tilde{\Theta}_1)\\\hat{f}(X,\dot{X}|\tilde{\Theta}_2)\\\hat{f}(X,\dot{X}|\tilde{\Theta}_3)\end{bmatrix}=\begin{bmatrix}\Theta_1^{\mathrm{T}}\varepsilon(X,\dot{X})\\\Theta_2^{\mathrm{T}}\varepsilon(X,\dot{X})\\\Theta_3^{\mathrm{T}}\varepsilon(X,\dot{X})\end{bmatrix} \tag{29}$$

Comparing (23) and (28), it can be observed that the fuzzy controller is a specific case of the proposed controller in this paper, in which the robust term $W = 0$. Moreover, the proposed control algorithm benefiting from internal force concept can ensure that all cables remain in tension while the end−grab moves along the designed pick−and−place trajectory. As a result, the proposed control scheme in this paper can bring a better comprehensive control performance.

Moreover, in order to have a quantitative evaluation and comparison between the performance of the two controllers, the root mean square error (RMSE) and maximum absolute of tracking errors (MAE) of the end−grab in all directions of the task space are presented, and they can be expressed as follows:

$$\mathrm{RMSE}=\sqrt{\frac{1}{N}\sum_{k=1}^{N}\left|e(k)\right|^2} \tag{30}$$

$$\mathrm{MAE}=\max\left(\left|e(k)\right|_{k=1\sim N}\right) \tag{31}$$

where $e(k)$ denotes the position tracking error of the end−grab; $N$ is the number of samples, and $k$ the current sample.

All dynamic and kinematic parameters of the cable−based gangue−sorting robot and the proposed controller parameters are given in Table 2. Without loss of generality, the lumped composite disturbance vector is set as $D=[4\sin(10t)\quad 2\sin(10t)\quad 4\sin(10t)]^{\mathrm{T}}$; the initial motion state of the end−grab is $X_0=[X_1,\dot{X}_1,X_2,\dot{X}_2,X_3,\dot{X}_3]=[1.3,\quad 0,\quad 1.2,\quad 0.03\pi,\quad 1.5,\quad 0]^{\mathrm{T}}$; the simulation time is set to 40 s; the membership function of the fuzzy control system is selected as $\mu_{A_i^l}(x_i)=\exp\left(\dfrac{-(x_i-\bar{x}_i^l)}{0.2}\right)^2$, in which $\bar{x}_i^l$ is 1, 2, 3, 4, and $A_i$ is NB, NS, ZO, PS, and PB, respectively.

**Table 2.** Dynamic and kinematic parameters of the robot and the proposed controller parameters.

| Parameter | Symbol | Value |
|---|---|---|
| Height of the pillar, Figure 3 | $h$ | 3 m |
| Length of the rectangle formed by $A_i$ | $b$ | 4 m |
| Width of the rectangle | a | 4 m |
| Mass of the end−grab | $m$ | 5 kg |
| Acceleration of gravity | $g$ | 9.8 m/s$^2$ |
| Gain matrix | $K_D$ | 250 $I_{4\times4}$ |
| Matrix of the sliding surface | $\varLambda$ | 10 $I_{3\times3}$ |
| Adaptation law matrix | $\boldsymbol{\Gamma}$ | Diag(10,10,10) × 10$^{-4}$ |
| Gain matrix of the robust term | $W$ | Diag(0.2,0.2,0.2) |

The spatial circle trajectory for the end−grab can be expressed as:

$$\begin{cases} x = 0.8\cos(0.1\pi t) + 1.8 \\ y = 0.8\sin(0.1\pi t) + 2 \\ z = 1.5 \end{cases} \tag{32}$$

The results of the position trajectory tracking of the end−grab are shown in Figure 10. It is observed that the end−grab tracks the planned spatial circle trajectory relatively well with the proposed robust adaptive fuzzy tracking controller. Figure 11 illustrates the position trajectory and the position errors of the end−grab in $x$−, $y$−, and $z$−directions, respectively. It can be seen that, compared with $x$−direction and $y$−directions, $z$−direction has the better tracking effect, and its absolute error is less than 6‰. Furthermore, the error in $z$−direction obviously changes in a fixed period, and the errors are always stable in the above range. Additionally, it is concluded from Figure 11d that the absolute value of the error in the $x$−direction and $y$−direction are all within 1%, fluctuating in the range of −8‰–(8‰). Moreover, the absolute value of the fluctuation is less than 3‰.

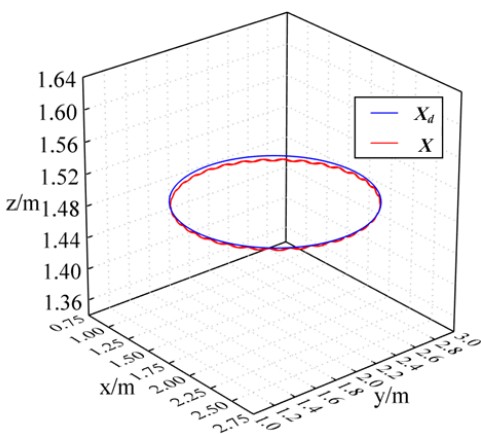

**Figure 10.** The spatial circle and trajectory tracking.

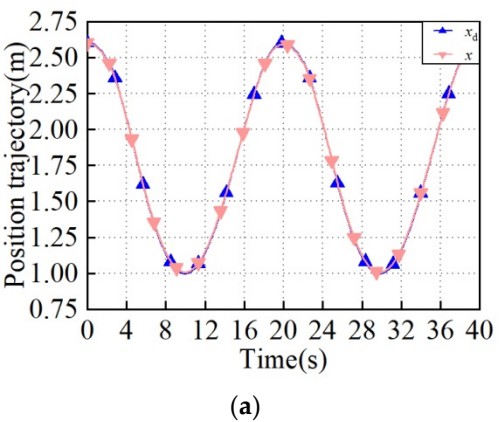

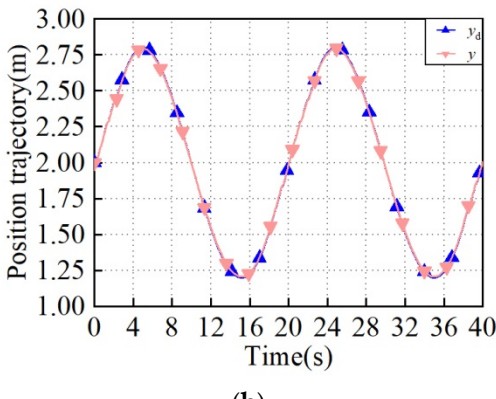

(**a**)

(**b**)

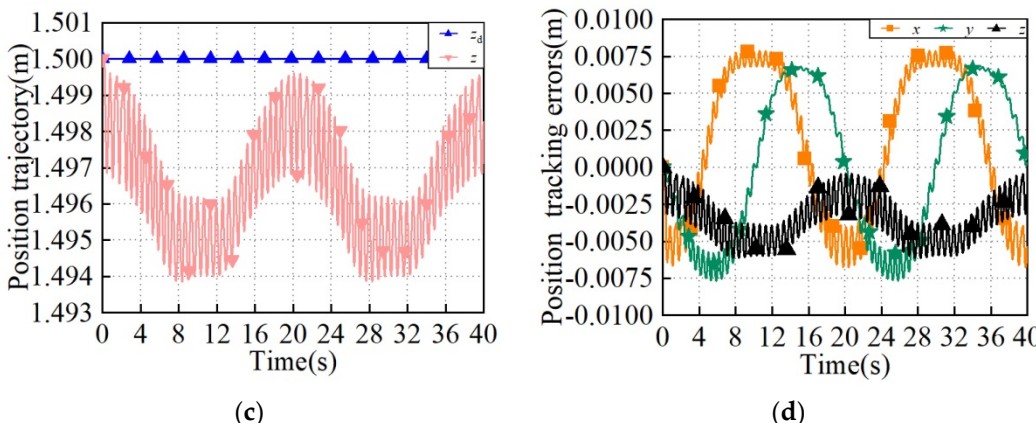

**Figure 11.** Position trajectory tracking of the end−grab for the spatial circle. (**a**) Position trajectory in *x*−direction; (**b**) position trajectory in *y*−direction; (**c**) position trajectory in *z*−direction; (**d**) trajectory tracking errors.

Figure 12 illustrates the velocity tracking and the velocity tracking errors of the end−grab in *x*−, *y*−, and *z*−directions, respectively. It can be observed that the absolute value of velocity error in *x*-direction is within 1.6%, and that in *y*-direction is within 1.5%, while that in *z*−direction is within 1.6%. Additionally, it can be seen that the velocities in the three directions fluctuate greatly within 0.5 s; however, the velocities in the three directions fluctuate steadily in an acceptable range.

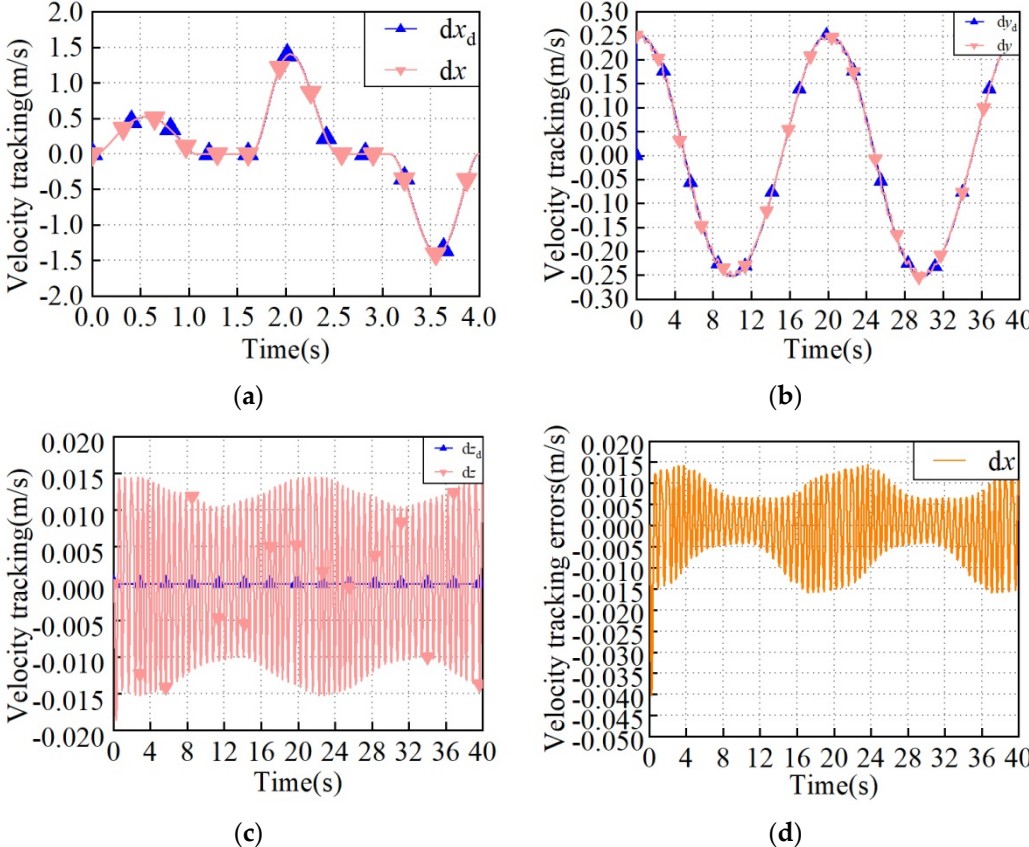

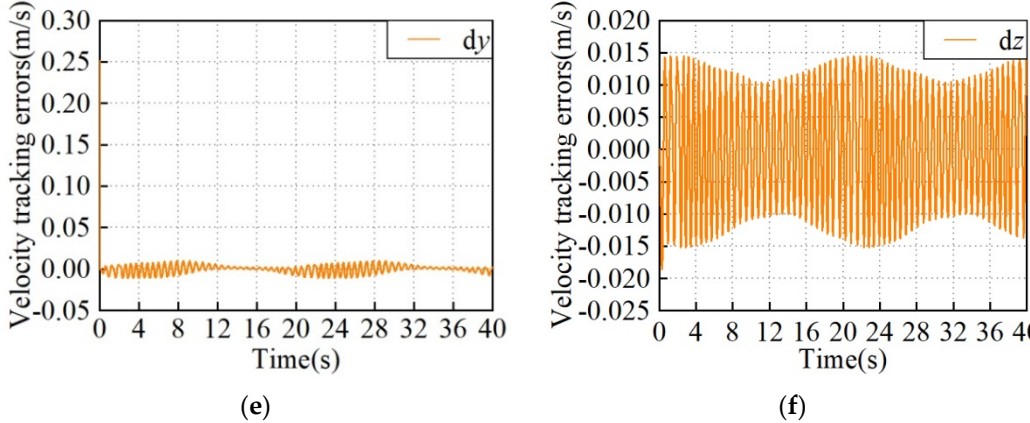

**Figure 12.** Velocity tracking of the end−grab for the spatial circle. (**a**) Velocity tracking in *x*−direction; (**b**) velocity tracking in *y*−direction; (**c**) velocity tracking in *z*−direction; (**d**) velocity tracking error in *x*−direction; (**e**) velocity tracking error in *y*−direction; (**f**) velocity tracking error in *z*−direction.

The four cable tensions while the end−grab follows the spatial circle defined by Equation (32) are shown in Figure 13. It is obvious that the tension of each cable is greater than 5 N, which satisfies the unidirectional characteristics of the cables. In addition, the phase, period, and fluctuation range of the cable tensions change steadily, which can generate smooth and continuous movement for the end−grab.

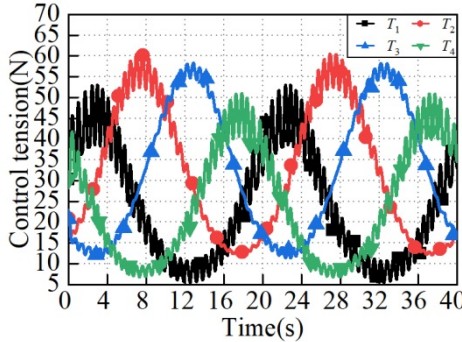

**Figure 13.** The control input cable tensions for the spatial circle.

While the initial motion state of the end−grab is set as $X_0=[X_1,\dot{X}_1,X_2,\dot{X}_2,X_3,\dot{X}_3]=[2.25,\ 0,\ 1,\ 0,\ 1.5,\ 0]^{\mathrm{T}}$, the simulation time is set to 4 s. The position tracking of the end−grab for the proposed pick−and−place trajectory is displayed in Figure 14. It is observed that the end−grab, in common with the spatial circle, tracks the proposed pick−and−place trajectory relatively well with the proposed robust adaptive fuzzy tracking controller.

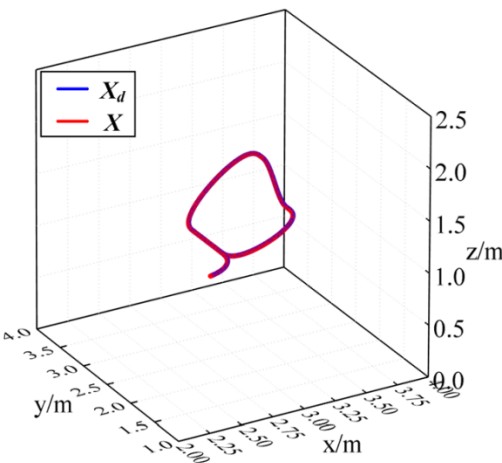

**Figure 14.** The proposed pick−and−place trajectory and trajectory tracking.

The position trajectory tracking errors of the end-effector in the $x$−direction, $y$−direction, and $z$−direction are shown in Figure 15. As can be seen from the figure, the absolute values of the errors in $x$−, $y$−, and $z$−directions are all within 2‰, and the error in $y$-direction is the largest. The maximum absolute value of the error in $y$−direction reaches 1.75‰ around 3.8 s. However, from the pick−and−place trajectory planning in Section 4.1, it can be seen that the end−grab has unloaded the gangues, which will not affect the smooth operation of the end−grab. Thus, the above error is acceptable.

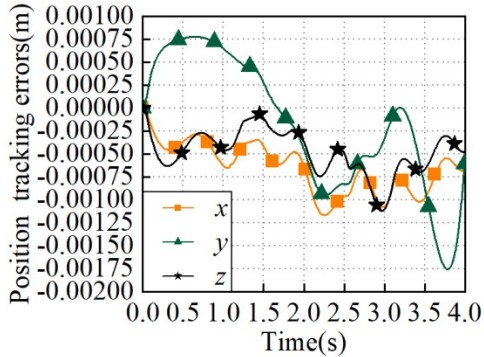

**Figure 15.** Position trajectory tracking error for the proposed pick−and−place trajectory.

RMSE and MAE for the both controllers in the task space while the end−grab moves along the proposed pick−and−place trajectory are listed in Table 3, respectively. The results obtained for the performance indices of the both controllers indicate the appropriate performance in practice. However, it can be seen from the qualitative analysis that the proposed controller has outperformed the fuzzy controller. As an example, the RMSE and MAE values of the proposed controller are $8.9867 \times 10^{-4}$ and $2 \times 10^{-2}$ m, respectively, which are better than that of the fuzzy controller, $9.1872 \times 10^{-4}$ and $2 \times 10^{-2}$. It can be seen from the MAE values of the two controllers that they are equal to each other, and this is because the maximum absolute of tracking errors occurs at the initial position of the pick−and−place trajectory for the end−grab. The initial motion state of the end−grab is set as equal to each other for the two controllers. It should be pointed out that the initial motion state of the end−grab has an effect on the RMSE and MAE.

**Table 3.** RMSE and MAE for the both controllers.

| Controller | RMSE | MAE |
|---|---|---|
| Proposed controller | $8.9867 \times 10^{-4}$ m | $2 \times 10^{-2}$ m |
| Fuzzy controller | $9.1872 \times 10^{-4}$ m | $2 \times 10^{-2}$ m |

Velocity tracking of the end−grab for the proposed pick−and−place trajectory is depicted in Figure 16. From Figure 16, we can see that the absolute values of velocity errors in $x-$, $y-$, and $z-$directions are within 1.8%, and thus, the velocity tracking effect is good. Furthermore, the velocity fluctuation is the most complicated from 1.8 s to 3.2 s. However, it can be seen from the figure that the absolute value of the maximum velocity fluctuation error is about 0.75%.

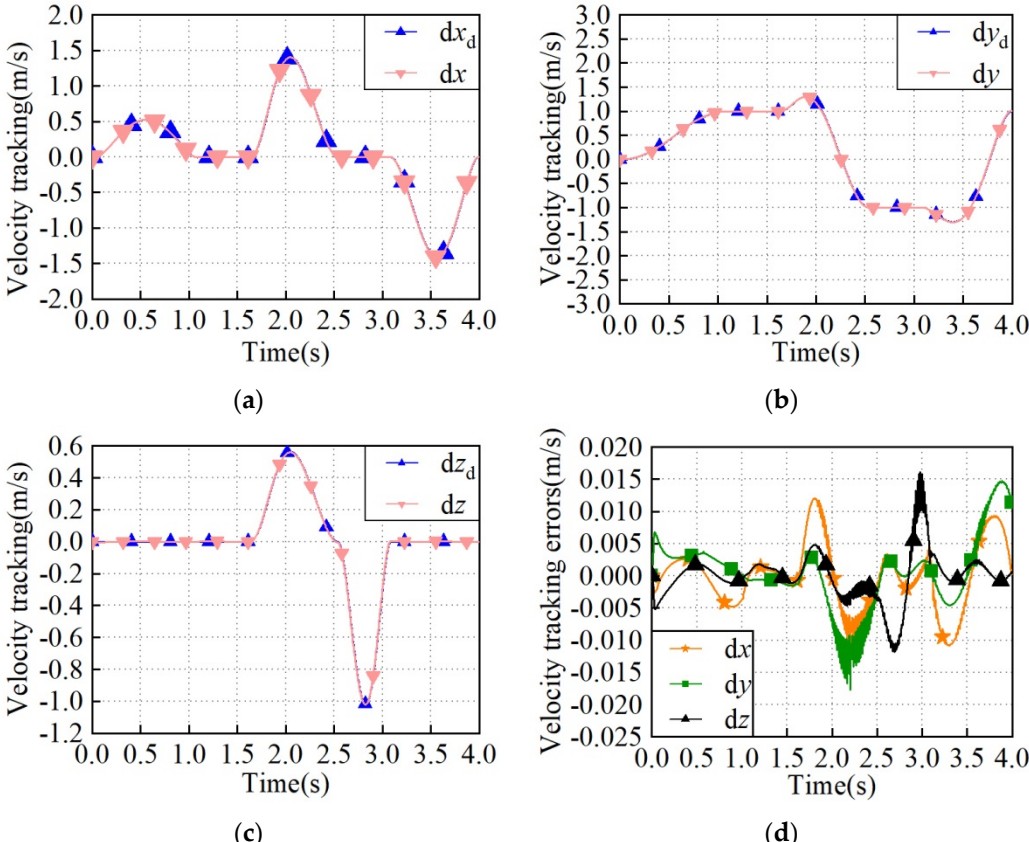

**Figure 16.** Velocity tracking of the end−grab for the proposed pick−and−place trajectory. (**a**) Velocity tracking in $x-$direction; (**b**) velocity tracking in $y-$direction; (**c**) velocity tracking in $z-$direction; (**d**) velocity tracking errors.

The four cable tensions while the end−grab follows the proposed pick−and−place trajectory are shown in Figure 17. It is obvious that the tension of each cable is also greater than 5 N, which satisfies the unidirectional characteristics of the cables. In addition, the cable tensions are smooth and continuous.

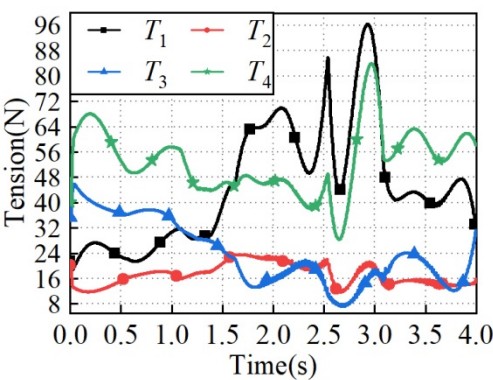

**Figure 17.** The control input cable tensions for the proposed pick−and−place trajectory.

The performed study clearly shows certain advantages of the proposed control system, which include: first, the presented four−phase pick−and−place trajectory planning scheme can generate a smooth and continuous trajectory for the end−grab to perform the pick−and−place operation of the MTGs; second, the proposed robust adaptive fuzzy tracking control strategy is able to track a trajectory as close as possible to the desired one without violating the cable tension limits.

## 5. Conclusions and Future Works

In this paper, a control system for the cable−based gangue−sorting robot performing the pick−and−place operation of MTGs with different shapes, sizes, and masses was presented, which consists of two modules: (i) the pick−and−place trajectory planning module and (ii) trajectory tracking control module.

In more detail, a four-stage trajectory planning scheme for the end−grab of the cable−based gangue−sorting robot is proposed in this paper. The pick−and−place trajectory of the end−grab is planned into four periods: the starting period, preparing period, picking period, and placing period. In addition, according to the established dynamic equation of the cable−based gangue−sorting robot, a robust adaptive fuzzy control strategy was designed to track a given trajectory in the presence of model uncertainties, varying payloads, and external disturbances. Based on Lyapunov stability analysis, the stability of the closed-loop control system was theoretically proved. To evaluate the proposed control system, numerical simulations were performed for the cable−based gangue−sorting robot. The simulation analysis of the pick−and−place trajectory planning scheme of the robot shows that the motion trajectory, velocity, and acceleration of the end−grab are smooth and continuous, and moreover, the cable length changes smoothly and continuously. Meanwhile, simulation analysis of the robust adaptive fuzzy control strategy of the robot shows that a high-precision position and velocity tracking performance for the end−grab was obtained with the proposed control strategy, which is robust against the uncertainties.

As a matter of fact, tracking, approaching, picking, and placing the MTGs have become an important mission of the cable−based gangue−sorting robots. In this paper, the authors present the proposed pick−and−place trajectory planning and trajectory tracking control for the robots. In the next phase, we will focus on the following parts as our future work: (i) contact and impact analysis during the picking and placing process of the MTGs; (ii) motion stabilization of the cable−based gangue−sorting robots after picking and placing the MTGs; and (iii) the experimental validation of the four-stage trajectory planning scheme and the proposed pick−and−place trajectory tracking control scheme for the cable−based gangue−sorting robots.

**Author Contributions:** Conceptualization, P.L. and L.G.; methodology, P.L.; software, Y.S.; validation, P.L. and X.Q.; formal analysis, Y.Q.; investigation, P.L.; resources, X.D.; data curation, X.C.; writing—original draft preparation, P.L.; writing—review and editing, P.L., Y.Q. and X.Q.; supervision, Y.Q.; project administration, X.C.; funding acquisition, P.L. and H.T. All authors have read and agreed to the published version of the manuscript.

**Funding:** This research was funded by the financial support of National Natural Science Foundation of China (NSFC) under Grant NO. 52174149, Shaanxi Province Natural Science Basic Research Program Project under Grant No. 2019JQ-796, and Key Research and Development Program of Shaanxi Province under Grant NO. 2022GY-241.

**Institutional Review Board Statement:** Not applicable.

**Informed Consent Statement:** Not applicable.

**Data Availability Statement:** The data used to support the findings of this study are available from the corresponding author upon request.

**Acknowledgments:** The research is supported by Open Fund of Key Laboratory of Electronic Equipment Structure Design (Ministry of Education) in Xidian University. And moreover, the authors are grateful to the guest editor, Zhufeng Shao, Dan Zhang, and Stéphane Caro as well as the anonymous reviewers for their constructive comments and helpful suggestions that greatly improved the quality of this article.

**Conflicts of Interest:** The authors declared no potential conflict of interest with respect to the research, authorship, and/or publication of this article.

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
