# Peer review of "Pick–and–Place Trajectory Planning and Robust Adaptive Fuzzy Tracking Control for Cable–Based Gangue–Sorting Robots with Model Uncertainties and External Disturbances"

_machines, doi:10.3390/machines10080714_

Round 1
Reviewer 1 Report
The paper presents and focuses more on the control strategy of the developed cable robot.
1. The results seem interesting but some experiments or ADAMS should be used to verify.
2. How to obtain stability workspace? Does the color mean something?
3. Sections 1.1 and 1.2 are too long and can be shortened.
Reviewer 2 Report
The paper presents a pick-and-place trajectory planning and control approach for cable-suspended parallel robots. Only numerical results are presented, and no comparisons with state-of-the-art works is shown in the manuscript.
The topics of the paper are interesting and suitable for the journal. However, the following points need to be clarified to improve the quality and the readability of the manuscript.
· The novelty of this paper with respect to the state of art is not clearly described. It is not easy to assess which is the main contribution of the work with respect to the present literature. I suggest including a bullet point list with the main novelties together with a table where the main differences, advantages and disadvantages of the work are compared with similar ones.
· The general validity of the results is difficult to assess. It would be suitable to test the performance of the proposed algorithm with multiple trials, e.g., with randomly generated pick and place points.
· The geometrical and dynamical parameters of the simulated robotic system should be included in the paper. Otherwise, it would be impossible for the reader to replicate the numerical results.
· The performance of the proposed approach should be compared with state-of-the-art similar works on the topic. Furthermore, it would be suitable to present the results also by means of quantitative indexes or metrics, and not only with graphical plots.
· The quality of the graphical plots should be improved. I suggest using different dash styles to improve the readability of the curves even in a grey scale printed paper.
· The literature review should be improved by considering additional works on the topic of modelling and control of cable-suspended parallel robots. Some suggested references are reported below.
[1] Scalera, L., Gasparetto, A., Zanotto, D. (2019). Design and experimental validation of a 3-DOF underactuated pendulum-like robot. IEEE/ASME Transactions on Mechatronics, 25(1), 217-228.
[2] Idá, E., Briot, S., Carricato, M. (2022). Identification of the inertial parameters of underactuated Cable-Driven Parallel Robots. Mechanism and Machine Theory, 167, 104504.
[3] Dion-Gauvin, P., Gosselin, C. (2022). Beyond-the-static-workspace point-to-point trajectory planning of a 6-DoF cable-suspended mechanism using oscillating SLERP. Mechanism and Machine Theory, 174, 104894.
Round 2
Reviewer 1 Report
The authors have revised the manuscript.
However, Section 1 is still long and adding more paragraphs wouldn't help to improve it.
Author Response
Point: Section 1 is still long and adding more paragraphs wouldn't help to improve it.
Response: We thank the reviewer for providing the constructive suggestion very much. Considering the reviewer’s suggestions, we have revised Sections 1 where some redundant information and sentences have been deleted in the revised manuscript. (page 1- page 4)
At last, special thanks for your good comments. We tried our best to improve this manuscript and made some changes in this manuscript. We appreciate for editors/reviewers’ warm work earnestly, and hope that the correction will meet with approval. Once again, thank you very much for your comments and suggestions.

Reviewer 2 Report
The paper has been improved with respect to the previous version.
Author Response
Thank you very much for your comments and suggestions.